# “It’s Just Not Something We Do at School”. Adolescent Boys’ Understanding, Perceptions, and Experiences of Muscular Fitness Activity

**DOI:** 10.3390/ijerph18094923

**Published:** 2021-05-05

**Authors:** Ashley Cox, Stuart J. Fairclough, Robert J. Noonan

**Affiliations:** 1Movement Behaviours, Health, and Wellbeing Research Group, Department of Sport and Physical Activity, Edge Hill University, Ormskirk L39, UK; faircls@edgehill.ac.uk; 2Appetite and Obesity Research Group, Department of Psychology, University of Liverpool, Liverpool L1, UK; r.noonan@liverpool.ac.uk

**Keywords:** muscular fitness, physical activity, adolescents, physical education

## Abstract

Background: English youth typically do not sufficiently engage in the types and intensities of physical activity that develop muscular fitness. The aim of this study was to use a combination of qualitative techniques to explore adolescent boys’ understanding, perceptions, and experiences of physical activity and the role muscular fitness plays within boys’ physically active lifestyles. Methods: Focus group interviews with a write, draw, show, and tell activity were conducted with 32 adolescent boys aged 14–16 years from 3 secondary schools. Three separate sources of data (frequency counts, verbatim transcripts, and visual data) were generated and were pooled together and triangulated. Data were analysed deductively, first using the Youth Physical Activity Promotion model as a thematic framework, and then inductively. Results: Physical activity was frequently associated with organised sport, and most boys were unaware of current UK physical activity guidelines. Co-participation was frequently reported as a reinforcing factor to physical activity. Conclusions: There was a perceived lack of opportunity to participate in muscular fitness activities, particularly in school, and knowledge of how to conduct muscular fitness activities was limited. The contribution of physical education was highlighted as being key to facilitating exposure to muscular fitness activities.

## 1. Introduction

Current physical activity guidelines for the UK and other developed countries suggest that children and young people should engage in a variety of types and intensities of physical activity across the week in order to develop movement skills, bone strength, and muscular fitness [1,2]. Despite the growing body of literature supporting the health benefits of muscular fitness [3,4,5], it is often an overlooked element of the physical activity guidelines, with much of the focus towards averaging at least 60 min of daily, aerobic-based, moderate-to-vigorous intensity physical activity over the week [6,7]. Throughout the last 50 years, muscular fitness levels have declined across most developed countries, including the UK [8]. Low muscular fitness contributes to the development of non-communicable disease risk in adolescents, leading to poor health in adulthood, including cardiovascular disease, osteoporosis, and type 2 diabetes [9,10,11]. Improving muscular fitness is associated with a range of health markers, including cardiorespiratory fitness [12,13], metabolic function [11,14,15], bone health [16]. and mental health [17] in youth.

Adolescent boys have been highlighted as a group that may respond well to muscular fitness activity [18], and recent evidence has confirmed the efficacy of muscular fitness activity when delivered in schools [19]. However, much of the evidence pertaining to muscular fitness amongst adolescents is quantitative in nature [20]. Formative research with the target group may overcome key intervention challenges, including recruitment and engagement, and could improve intervention efficacy [21,22]. Adolescents have previously suggested that barriers to conducting physical activity include lack of facilities, high costs, and accessibility of facilities in which to conduct physical activity [21,22,23,24,25]; however, none of these studies address muscular fitness activity independently. Moreover, adolescence is key to developing healthy habits and behaviours that carry over into adulthood [26]. Therefore, if future research is to implement muscular fitness activity interventions in adolescent boys, with schools as the predominant setting, then contextualising boys’ understanding, perceptions, and experiences of muscular fitness activity may support intervention success.

The use of qualitative data can provide additional context to key factors that facilitate and hinder adolescents’ participation in physical activities [27]. Socio-ecological models provide a framework through which to understand the various personal, social, and environmental factors that facilitate and restrict adolescent physical activity [28,29,30]. These factors are represented in the Youth Physical Activity Promotion model [30]. The Youth Physical Activity Promotion model’s socio-ecological approach provides a recognised framework that goes beyond investigating the individual factors associated with physical activity, allowing for the social and constructed environments to be considered and inform future intervention designs [31]. To date, much of the qualitative investigation into adolescent physical activity has relied on focus group interviews alone [21,32,33]. However, adolescents are at a vulnerable stage in their development, and often have difficulty in expressing their thoughts, feelings, and emotions verbally [34]. Therefore, qualitative investigation within this population group requires careful consideration of the methodologies used, and may benefit from the use of visual methods to engage participants [34,35]. Focus group interviews in combination with drawings may allow participants to express themselves more effectively than when using verbal communication alone. Several research studies have utilised drawing techniques within focus group interviews [36], in order to engage adolescents in the research process and facilitate communication and discussion around health-related topics [37,38,39]. The “write, draw, show, and tell” method provides participants with alternative ways of expression, thereby enabling a deeper exploration of their perceptions and experiences by not limiting participants to verbal communication alone. The multiple-methods-based approach may cultivate greater inclusivity, and elicit more representative and complete perceptions on physical activity topics, when compared to singular-methods-based approaches [40].

The aim of this study was to use a combination of qualitative techniques to explore adolescent boys’ understanding, perceptions, and experiences of physical activity, and the role muscular fitness plays within boys’ physically active lifestyles. It was envisaged that the contextual information gathered from this study will provide (a) novel insights into the meanings adolescent boys ascribe towards physical activity, and (b) inform the design of future physical activity promotion strategies targeting adolescent boys.

## 2. Materials and Methods

### 2.1. Participants

To ensure explicit and comprehensive reporting, the consolidated criteria for reporting qualitative research were used to guide the reporting process [41]. Eight schools situated in a range of deprivation areas were provided with study information and invited to participate in the study. Three schools agreed to participate in the study, with data collection taking place during mandatory physical education classes, which provided a mixed physical ability sample. All eligible boys had taken part in a previous cross-sectional study, and completed a study pack containing parent/carer and child information sheets, parent/carer consent forms, and participant assent forms. For the purposes of this study, 32 consenting adolescent boys (aged 15.23 ± 0.60 years) from across three schools in North West England were randomly selected to take part via a lottery method. Ethical approval for this study was granted by the Edge Hill University Ethics Committee (SPA-REC-2017-321), and data collection took place between November 2019 and January 2020.

### 2.2. Procedures

The research took a phenomenological approach. Five focus group interviews with a write, draw, show, and tell activity were arranged and conducted by the first author at school sites during timetabled physical education lessons. Each focus group interview comprised between five and eight (seven, five, five, eight, and seven, respectively) participants, and was conducted between 09:30 a.m. and 01:00 p.m. so as to limit participant fatigue and restlessness [42]. Other than participant age, there were no specific inclusion criteria employed. In order to maintain the interest and enthusiasm of participants and accommodate short attention spans [43,44,45], each focus group interview comprised a range of different interactive activities and lasted between 19 and 21 (mean = 20.11) min. The 20 min time allocation was stipulated by the first participating school, and was utilised in subsequent schools in order to minimise bias and ensure consistency and standardisation across other schools.

Semi-structured focus group interview guides were used to ensure consistency across each focus group interview. The semi-structured focus group interview guides were informed and structured around the multiple-methods write, draw, show, and tell framework [40] (see [33,46,47] and Table 1 for details). The multiple-methods approach was incorporated into the focus group interviews in order to further stimulate participants’ thinking and facilitate discussion around physical activity. The write, draw, show, and tell framework has been used in previous qualitative research investigating young people’s perceptions of physical activity [40,47]. Focus group interview questions were underpinned by the Youth Physical Activity Promotion model, which acknowledges the various individual, social, and environmental factors that influence physical activity [30]. Example write, draw, show, and tell questions used throughout the focus group interviews are presented in Table 1. The Youth Physical Activity Promotion model describes three factors that predispose, reinforce, or enable physical activity behaviour. Predisposing factors include variables that increase the likelihood of engaging in physical activity, and are based upon a self-evaluative construct that reduces physical activity behaviour into two fundamental questions: “Am I able?” and “Is it worth it?” These two fundamental questions explore attitudes and beliefs about physical activity and perceptions of self-confidence and self-worth. Reinforcing factors include variables that reinforce physical activity behaviour, and may include peers, family, coaches, and teachers. Finally, enabling factors include variables that allow youth to be physically active, and may consist of environmental and biological factors.

Focus group interviews were conducted in quiet, non-intrusive school classrooms, where participants and researcher alike could be overlooked but not overheard. A circular seating arrangement was used, with the researcher sat with the children, and the researcher and children addressed one another by their first names [48]. Following explanation of the procedure, verbal consent was obtained from all participants. During the focus group interviews, efforts were made by the first author to ensure that all individuals participated—for example, by making eye contact with participants to encourage them to contribute to the discussion, and by asking individuals to expand on their individual responses [48].

An icebreaker activity conducted at the beginning of each focus group interview provided participants the opportunity to experience speaking aloud, and helped to establish an environment where individual contributions were welcomed, encouraged, and respected. Participants were provided with Post-It note© (3M, Bracknell, UK) paper and asked to write down ”five words to best describe physical activity to someone else”. Participants then placed their responses on a centralised whiteboard and were asked to provide further detail on their response. Following the Post-It note task, participants were provided with a series of open-ended questions aligned to the aims of the study (see Table 1). In order to improve the flow of discussion and engage all participants, open questions starting with ”what” and ”how” were posed to participants. Participants were then provided with the opportunity to discuss topics amongst themselves, with the lead author repeating back responses on their main discussions in order to ensure correct interpretation and gain clarity regarding group discussions, and to further engage shy participants [49]. This method of respondent validation has been used elsewhere with this age group, and helps increase the legitimacy and trustworthiness of the reported findings [21,50].

In order to further triangulate the data and ensure the credibility and dependability of the findings, participants were invited to express their perceptions of physical activity through a write/draw activity. Participants were asked to independently “draw an area, place, space, or environment where you are most likely to be active”. The drawing task took the focus away from direct questioning and consensus, allowing participants to contribute and engage through other means, and thus strengthen the study’s findings [51]. Throughout the draw activity the first author separately engaged children in informal conversations in order for them to articulate what they were drawing, and why. All focus group interview discussions were audio recorded and transcribed verbatim.

### 2.3. Data Management and Analysis

The focus group interviews provided three sources of data: frequency count (Post-It note©; show activity), visual data (write and draw activity), and verbatim data (tell activity and participants’ write and draw narratives). Frequency counts, visual data, and verbatim data were then pooled together in order to explore and expand upon emergent themes and clarify findings. This multifaceted data collection approach allowed for data triangulation, which minimised misinterpretation and, in doing so, enhanced the credibility of the findings [35,52]. All data were managed in NVivo12 (Version 12.6.0; QSR International Pty Ltd, Victoria, Australia) and analysed independently. Data were first analysed deductively, using the Youth Physical Activity Promotion model as a thematic framework, and then inductively, in order to enable emergent themes to be further explored. Braun and Clarke’s phases of thematic analysis were used to inform the coding process once the lead author had become familiar with the transcripts [53]. This approach to analysis allowed for flexibility, and helped to examine the different perspectives of the participants and summarise the key points of a large data set [54]. A set of codes were generated, defined, and named that aligned with the write, draw, show, and tell questions, the Youth Physical Activity Promotion model, and relevant themes that provide insight into the understanding, perceptions, and experiences of muscular fitness activity among adolescent boys. The third author then reviewed the coding process and provided suggestions to ensure that the coding was representative of the YPAPM and existing literature terminology. Further researcher triangulation took the form of a presentation of the verbatim quotations and drawings to the third author as a critical friend who had previously independently reviewed the data sources and cross-examined the data sources against the themes in reverse in order to offer alternative perspectives. This process was repeated until a minimum 90% agreement level had been reached by the two authors [55,56,57]. Once both authors were in agreement, the final codes were counted and placed into pen profiles for reporting. The pen profile approach has been used in previous physical activity research with young people and adolescents [21,22,24,40]. Quotations from participants were used to demonstrate key points and discussions under each theme, and then placed into pen profiles in order to provide further context and insight to the discussions.

## 3. Results

### 3.1. Ice Breaker Results

A total of 135 responses were recorded for the icebreaker activity/post it note task. Physical activity was most frequently associated with fun (*n* = 18), enjoyable (*n* = 11), and healthy (*n* = 9); the top 10 responses are presented in Table 2.

### 3.2. Knowledge of Physical Activity

Participants’ knowledge of physical activity was placed into one of three coding categories, and is presented in Figure 1. Three themes of physical activity knowledge, no knowledge, and incorrect interpretation of physical activity guidelines were further subdivided into aerobic and muscular fitness activity, in order to conduct further inductive content analysis and to establish areas of physical activity knowledge. Participants were deemed to incorrectly interpret physical activity guidelines if they could not accurately relay what the current recommendations were.

### 3.3. Predisposing Factors to Physical Activity

Profiles representing boys’ perceived predisposing factors to conducting physical activity are displayed in Figure 2, with the two fundamental questions of “Am I able?” and “Is it worth it?” utilised to perform coding [31]. Five subthemes of competence +ve (*n* = 12), competence −ve (*n* = 6), masculinity +ve (*n* = 6), enjoyment +ve (*n* = 8), and enjoyment −ve (*n* = 3) were linked to predisposing factors to conducting physical activity. Both positive (+ve) and negative (−ve) influences featured in primary predisposing themes.

### 3.4. Reinforcing Factors to Physical Activity

Boys’ perceived reinforcing factors to physical activity are presented in Figure 3, with six primary themes: parental support, parent attitudes/beliefs, teacher support, sibling support, community, and friends. A further nine secondary themes were identified: interest −ve (*n* = 1), broken family −ve (*n* = 1), injury concerns −ve (*n* = 2), logistical support +ve (*n* = 2), sibling co-participation +ve (*n* = 4), knowledge development −ve (*n* = 14), co-participation (teachers) +ve (*n* = 4), co-participation (friends) −ve (*n* = 3), and co-participation (friends) +ve (*n* = 11). Positive (+ve) and negative (−ve) influences featured in both primary and secondary reinforcing themes.

### 3.5. Enabling Factors to Physical Activity

Boys’ perceived enabling factors to physical activity are presented in Figure 4. There were four primary themes: environment, fitness, time, and technology; and six secondary themes: crime −ve (*n* = 4), provision −ve (*n* = 15), provision +ve (*n* = 8), proximity −ve (*n* = 8), proximity +ve (*n* = 7), and traffic −ve (*n* = 2). Positive (+ve) and negative (−ve) influences featured in both primary and secondary enabling themes.

### 3.6. Write and Draw

Thirty-two boys completed the write and draw task. There were 111 marks from reports on specific themes. Figure 5 illustrates the emerging themes with five primary themes: activity, co-participation, parental support, coach support, and physical environment. A further 12 secondary themes were identified: siblings, friends, logistic support, verbal encouragement, home, school, sports field, gym, sports facility or leisure centre, muscular fitness activity, aerobic-based physical activity, and traditional team sport.

## 4. Discussion

The aim of this study was to use a combination of qualitative techniques to explore adolescent boys’ understanding, perceptions, and experiences of physical activity, and the role muscular fitness plays within boys’ physically active lifestyles. It was envisaged that the contextual information gathered from this study would provide (a) novel insights into the meanings adolescent boys ascribe towards physical activity, and (b) inform the design of future physical activity promotion strategies targeting adolescent boys. Additionally, by investigating the perceptions and experiences of muscular fitness activity through discussing physical activity as a whole, we have minimised the potential for unintentional bias that may arise from discussing muscular fitness activity alone.

### 4.1. Physical Activity Knowledge

Physical activity guideline knowledge in adolescents has previously focused on the moderate-to-vigorous physical activity aspect of the current guidelines [58]. Limited knowledge of physical activity guidelines in adulthood has been reported, and suggests that a lack of knowledge may impact motivation to meet the suggested physical activity recommendations to benefit health [59,60]. According to the “Knowledge, Attitude, and Practices” model, individuals may modify their health and lifestyle behaviours if they are provided with specific knowledge to act upon. Our findings suggest an incorrect interpretation and a lack of knowledge surrounding physical activity guidelines amongst adolescent boys, particularly around muscular fitness activity. Given the declines in muscular fitness reported in many developed countries, there may be a gap in the dissemination of physical activity knowledge during a period in time when adolescents begin to form their own attitudes, beliefs, and behavioural habits, which are carried through into adulthood [26,61]. A lack of exposure to muscular fitness activity is likely to result in a limited understanding of muscular fitness activity, which was evident in the present study both in adolescent boys’ narratives and in their drawings. For example:

*“I wouldn’t know. Not been told about it [muscular fitness] really.”* (ID32).

The school environment has been shown to be effective in the promotion of physical activity in adolescents [62]. Interestingly, adolescents in this study considered the school to be the sole facilitator of physical activity knowledge acquisition. Additionally, the school environment provides access to muscular fitness activity independent of a pupil’s background and socio-economic status [63]. This may expose adolescents to varying forms of physical activity, including muscular fitness activity, and may enhance their knowledge and understanding of muscular fitness activity. However, our findings suggest that there is a perceived lack of understanding of, and exposure to, multiple forms of physical activity in schools, despite the need for exposure being highlighted within the national curriculum [64]. It has been suggested that the school environment and physical education in particular are well placed to enhance young people’s knowledge of lifelong health [65]. Therefore, exposure and teaching specific to physical activity guidelines designed to support health may cultivate an awareness of healthy behaviours as adolescents transition into adulthood. Ensuring the acquisition of knowledge regarding muscular fitness activity may support lifelong engagement in a popular mode of adult physical activity, and work towards reducing the lack of physical activity knowledge evidenced in adults [59]. Unfortunately, the participants in this study reported that the development of knowledge surrounding muscular fitness was rarely addressed by teachers.

### 4.2. Predisposing Factors

It is well accepted that physical activity enjoyment contributes to adolescent physical activity engagement [66]. Adolescents within this study associated physical activity with fun and enjoyment, whilst remaining cognisant of the associated health benefits. Furthermore, this study demonstrated that adolescent boys perceive muscular fitness activity as representing a masculine stereotype, and being an attractive form of physical activity. This is consistent with other studies [66,67], and further supports the potential role muscular fitness activity has for increasing overall physical activity in adolescent boys. In this study, perceived competence influenced participation both positively and negatively. For example,

*“It’s quite, like, rewarding if you do something well in football and because we do it all the time we get better.” (ID08) and “No. I am not very good at sports and that. I did American football at a summer camp once, but here (school) only does the main sports, like football.”* (ID21).

Predisposing factors to physical activity reported in this study were predominantly based on traditional team sport competence. Adolescents require exposure to a variety of physical activities in order to help them identify a form of physical activity they enjoy, and support lifelong physical activity participation [66]. Although traditional team sports remain popular during childhood, and dominate the physical education curriculum, participation in gym-based and less formal fitness activities tends to increase through adolescence [68]. Furthermore, during adulthood, fitness and gym-based activities are favoured over traditional team sports [69], and a lack of exposure during formative adolescent years may result in a lack of competence and enjoyment in adulthood. Given that musculoskeletal issues are the greatest cause of work sickness absence, and a primary cause of disability and loss of independence during adulthood [3,70], efforts should be made to engage adolescents in muscular fitness activity and cultivate a sense of competence. When adolescents spoke about perceived competence specific to muscular fitness activity, they expressed that they were not old enough to conduct muscular fitness activity, and suggested that they were only able to conduct team sports safely. This is a misconception that may impact predisposing factors to conducting muscular fitness activity, and provides an area worthy of further investigation.

The “Am I able?” construct of the Youth Physical Activity Promotion model is operationalised as the individual’s perception of competence in conducting physical activity [71]. To date, much of the literature has focused on aerobic, moderate-to-vigorous physical activity, and reports the effects of school-based interventions on moderate-to-vigorous physical activity as being non-existent and non-significant [63,72]. Moreover, it is recognised that physical education plays an important role in developing competence [73,74], yet traditional team sports and increasing aerobic moderate-to-vigorous physical activity may not engage the least active and least skilled adolescents, who would benefit the most from improved perceived competence [67,75]. Audio and visual data captured in this study suggest that adolescent boys may not be routinely exposed to muscular fitness activity in school. However, the implementation of muscular fitness in a school setting may enhance the perceptions of competence in adolescents who are less active and less skilled at traditional forms of physical education, by providing them with an alternative form of physical activity—especially in those who are overweight or obese [76]. Indeed, the potential for improving “the self” (e.g., self-esteem, self-efficacy, self-perceptions, etc.) through muscular fitness activity has been evidenced in recent research [77]. Schools provide a unique environment for regular, structured engagement in muscular fitness activity that can help adolescents to develop the skills, knowledge, and confidence to conduct muscular fitness activity safely and effectively [67,78]. The development of the knowledge, skills, and confidence to conduct muscular fitness may satisfy the “Am I able?” construct of the Youth Physical Activity Promotion model.

### 4.3. Reinforcing Factors

Consistent with prior research, friends provided social support in the form of co-participation (i.e., engaging in activity together [79,80,81]. It has been acknowledged that friend co-participation becomes more salient and critical in adolescents with respect to attitudes, activity decision-making, and emotional well-being [82]. Co-participation can influence physical activity by providing social support and establishing social norms that constrain or enable health promoting behaviours [79,81,83]. Furthermore, throughout adolescence, time spent with friends increases when compared to time spent with parents [84,85]. Although the influence of friends on physical activity is acknowledged, much of the work conducted to date focuses on adolescent girls’ relationships [86,87]. It has been suggested that interventions to increase physical activity, including muscular fitness activity, should provide adolescents with the skills to maintain and develop social networks that support participation [88]. However, the findings from this study suggest that adolescents do not view muscular fitness activity as a social activity, often associating the important social contribution with traditional team sports. Furthermore, only one (Figure 6) write, draw, show, and tell activity provided an example of muscular fitness activity and the potential for co-participation, but this was perceived as only being possible outside of school.

Given the importance of social networks in physical activity during adolescence, there is a requirement to provide opportunities to conduct muscular fitness activity in a social environment. Additionally, although previous research supports the positive role of friend co-participation [83], it generally focuses on traditional aerobic physical activity [58,81,89,90,91]. Given that team sport participation decreases and gym-based muscular fitness activity increases during adulthood [69], it may be of benefit to provide adolescents with a social environment to align their behaviour with the norms of their group, or the group that they want to belong to [92]. Indeed, it is acknowledged that schools play a key role in the social development of adolescents, and provide an environment in which adolescents can develop friend-to-friend with the support of teachers [93]. During adolescence, the sibling influence on physical activity as a family member may be tempered, but the influence of a sibling may contribute to healthier physical activity patterns [94]. Interestingly, activity specific to developing muscular fitness was more frequently associated with older sibling co-participation. For example,

*“Been to the gym with my sister before like. The first time that she went, lifting weight and that.” (ID26) and “Yes. My brother is a PT so I kind of get it. He shows me and like, has his insta page you can follow.”* (ID28)

Previous research has focussed predominantly on the relationship between siblings, moderate-to-vigorous physical activity, and team sport elements of physical activity [95]. Our findings suggest that sibling co-participation may support involvement in muscular fitness activity. However, siblings reinforcing participation in muscular fitness activity in this study were older, and took the role of a teacher and mentor in muscular fitness activity participation. This finding raises concerns over quality of provision and subsequent involvement in muscular fitness activity from the adolescents involved in this study. It is acknowledged that developmental differences and sibling rivalry may have a negative impact on physical activity participation [95,96]. Given that the development of muscular fitness in upper and lower limbs is not homogeneous, and may vary throughout growth and maturation [97,98,99], caution must be exercised by supportive older siblings in order to avoid the risk of injury through overexertion. Further research exploring the role of sibling co-participation in the development of muscular fitness is required.

Relationships between adolescents and their parents evolve from those established during childhood, with adolescents becoming more independent from their parents over time [79]. Within our focus group interviews it was apparent that parents had a role in supporting and influencing physical activity. However, adolescents in this study perceived their parents as having no interest or concerns regarding injury when discussing attitudes and beliefs towards muscular fitness activity, particularly when focusing on traditional forms of muscular fitness, such as weightlifting. These concerns suggest that the knowledge of parents regarding the contribution muscular fitness activity has to their children’s health is limited. Improving parental knowledge regarding the benefits of muscular fitness activity and dispelling unfounded safety concerns may support successful future interventions. Indeed, parent involvement in the design and subsequent implementation of school-based physical activity interventions has been welcomed by parents [100]. However, there is little research on how parents can help support the implementation of school-based muscular fitness interventions, suggesting that further work in this area is required. 

Despite the role teachers have in promoting muscular fitness activity, adolescents highlighted teachers as having a negative influence on their muscular fitness activity. For example,

*“I mean, we’ve been told [by teachers] it [muscular fitness] can be dangerous. They have a gym here [school], but they [teachers] don’t really use it that much. The main thing is like sports and that.”* (ID31).

Furthermore, it is suggested that teachers may approach muscular fitness activity with trepidation due to outdated misconceptions regarding the risk of injury, highlighting a need for continued professional development for physical education teachers to prescribe muscular fitness activity. Moreover, the risk-averse approach to school-based muscular fitness activity is further compounded by fears of litigation, despite governing bodies encouraging exposure to a wide range of activities provided the appropriate measures have been put in place [101]. Adolescents within this study felt as though teachers did not want to conduct muscular fitness activity, and provided limited choices of physical activities during timetabled physical education. Some adolescents ascribed this to a lack of willingness, although international data suggest that teachers rate muscular fitness activity as a priority in physical education [102,103,104]. Further investigation into the thoughts, experiences, and perceptions of muscular fitness activity amongst teachers is required in order to ensure successful implementation of school-based muscular fitness activity.

### 4.4. Enabling Factors

Enabling factors include variables that allow adolescents to conduct physical activity [30]. Such factors include equipment, access to parks or facilities, and the school environment. Visual and narrative data generated in this study further highlighted the importance of the school environment in supporting physical activity. 

During this study schools were suggested to be environments where physical activity was predominantly traditional team-sport-focused and muscular fitness activity was reserved for out of school (see Figure 7). Unfortunately, adolescents did not attribute the school environment to enabling participation to muscular fitness. This supports the notion that muscular fitness development is not catered for adequately in physical education, and in the school setting in general [105]. Recent literature suggests that the lack of implementation of muscular fitness activity in physical education and the school setting may be due to a lack of pedagogical understanding [103], and further investigation is required in order to understand the barriers to and facilitators of schools implementing muscular fitness activity.

The majority of adolescents in this study reported provision as a key barrier to out-of-school muscular fitness activity. Although the authors acknowledge that the costs of physical activity provision will be determined in part by the physical configuration of the localities in which individuals live, the findings of this study demonstrate the importance of the school setting in providing access to muscular fitness activity. Reductions in local authority budgets across the UK have led to cuts across various municipal sports facilities and leisure centres [106,107]. Cuts to the provision of local authority leisure facilities were reported within the focus group interviews, with local authority leisure centre closures—and a resultant overreliance on grassroots sports—presented as barriers to muscular fitness activity participation. For example,

*“I like, I don’t really do much in the way of like actual sport more like the gym. But, like, I haven’t been for a bit, because they’ve (local authority leisure centre) been closed.” (ID08) and “I don’t do it outside school. Nowhere does it, all mostly football.” (ID25) and ‘Yeah, council ones get ruined.”* (ID21).

Although access to a gym or leisure facility is not necessary for bodyweight muscular fitness activity, it may not expose adolescents to the full spectrum of muscular fitness activity that supports strength, power, and hypertrophy development [108]. Moreover, recent data suggest that bodyweight movements to develop muscular fitness may not be as effective as traditional forms of muscular fitness activity, which would require access to a gym or facility [19]. Furthermore, the associated costs of accessing a commercial gym resulted in shared memberships from adolescents interviewed in this study. For example,

*“That’s what I mean, it’s just better to go halves and that. It’s only a pin and you don’t get asked. I don’t think we’re supposed to be there either really, just with being young.”* (ID11).

The costs associated with accessing commercial and local authority facilities may prevent individuals from socio-economically disadvantaged groups from accessing muscular fitness activity, and further widen health inequalities. It has been suggested that removing user charges from accessing leisure facilities can increase overall physical activity and reduce health inequalities [107]. To date, this has not been explored with regards to muscular fitness activity in adolescents specifically. However, given the interest expressed amongst adolescents in participating in muscular fitness activity, future research may benefit from investigating the feasibility of providing reduced charges to adolescents to access both commercial and local authority facilities. Lack of provision and access to muscular fitness activity outside of school is further compounded by age restrictions put in place by commercial and [109] and local authority facilities [110]. Future research should explore how commercial and local authority facilities can better cater for adolescents in order to enable muscular fitness activity outside of school and support their transition into adulthood. 

### 4.5. Limitations and Recommendations

There are some limitations to this study that should be considered when interpreting its results. Although this study utilised a write, draw, show, and tell methodology, this has not been widely used in adolescent populations. However, the participants in this study responded well to this combination of interactive tasks, which provides evidence for future adolescent studies to adopt a similar methodological approach. The dual-methods approach provided the participants with alternative ways of expression, which not only fostered greater inclusivity, but also allowed for a deeper exploration of perceptions and experiences by not limiting participants to verbal communication alone. In doing so, the combination of methods revealed interconnected and complementary findings, which enhanced data credibility. However, due to curriculum commitments placed upon the adolescents involved in this study, a 20 min focus group interview length was stipulated by participating schools, and was used across all focus group interviews for standardisation. Despite this limitation, the results of this study offer novel insights into the experiences, understanding, and perceptions of physical activity and muscular fitness in adolescent boys, and the 20 min focus group interview length has been used elsewhere in the literature [48]. 

Given the contribution physical education provides to overall physical activity, further investigation into muscular fitness provision in school is required. Future research should investigate the knowledge physical education teachers have regarding muscular fitness, and investigate how current knowledge levels influence practice. Furthermore, an understanding of school policy and access to muscular fitness activity in schools may provide insight into the feasibility of muscular fitness activity provision. Finally, our findings suggest that friends are a key reinforcing factor to participating in physical activity. Previous research in adolescent females suggests that peer-led physical activity interventions may be effective [111,112,113]. However, there is a lack of peer-led physical activity interventions in adolescent boys. Given the importance of co-participation with friends, further research into expanding opportunities to conduct muscular fitness in school and out of school is required. Such research should be conducted with schools, government, and commercial gym facilities in order to provide opportunities for adolescents to conduct muscular fitness activity as they transition into adulthood.

## 5. Conclusions

Our results demonstrate a lack of knowledge surrounding physical activity guidelines amongst adolescent boys, particularly around muscular fitness activity. A desire to demonstrate a level of competency in activities that are deemed masculine may be satisfied through the delivery of muscular fitness activity, and may appeal to adolescent boys as an appealing form of physical activity. Despite the importance of muscular fitness in the healthy development of adolescents, there is a perceived lack of opportunities to participate in muscular fitness activity both in and out of school. The contribution of physical education was highlighted as being key to facilitating exposure to muscular fitness activity. Therefore, physical education programmes should ensure opportunities for muscular fitness development through engagement in developmentally appropriate activities. Furthermore, our findings suggest that co-participation with friends is a key reinforcing factor to conducting physical activity, yet opportunities to co-participate in muscular fitness activity are seldom, particularly at school. The significant financial costs and age restrictions associated with commercial gym use and memberships were seen as a barrier to out-of-school muscular fitness activity, reinforcing the need for exposure to this important mode of physical activity in schools.

## Figures and Tables

**Figure 1 ijerph-18-04923-f001:**
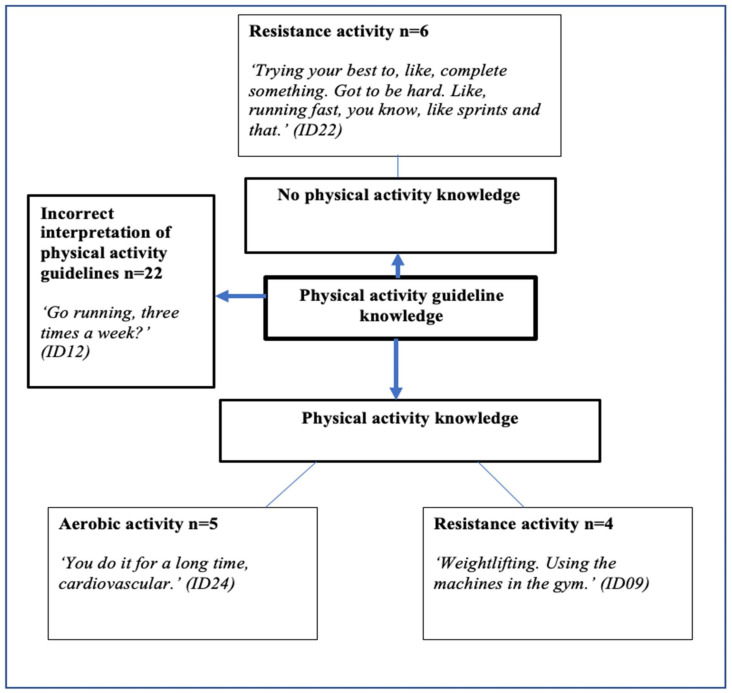
Physical activity guideline knowledge. +ve = positive; −ve = negative.

**Figure 2 ijerph-18-04923-f002:**
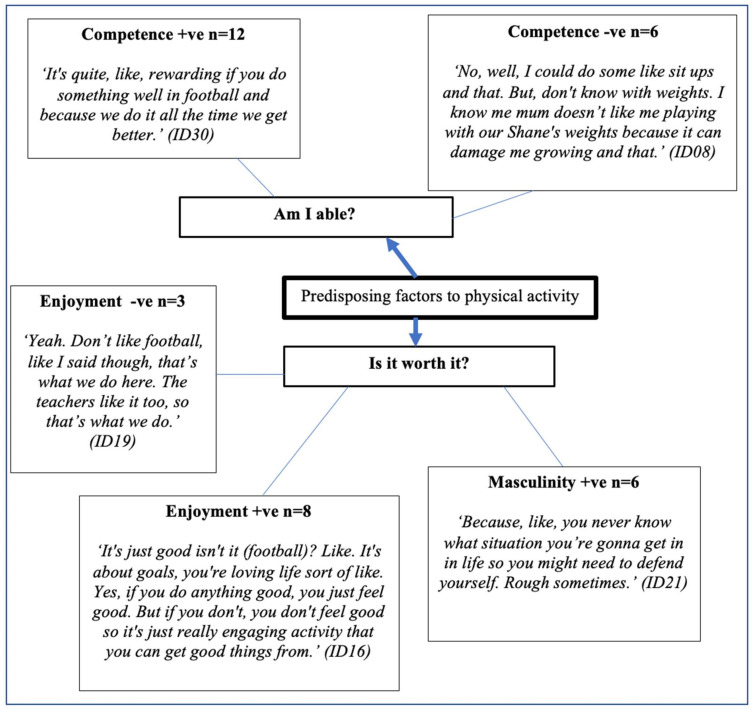
Predisposing factors to physical activity. +ve = positive; −ve = negative.

**Figure 3 ijerph-18-04923-f003:**
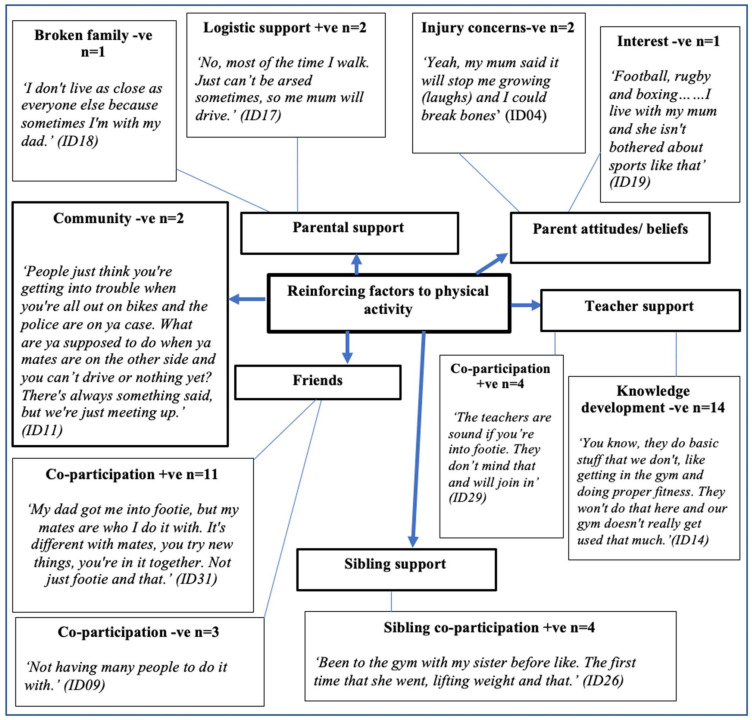
Reinforcing factors to physical activity. +ve = positive; −ve = negative.

**Figure 4 ijerph-18-04923-f004:**
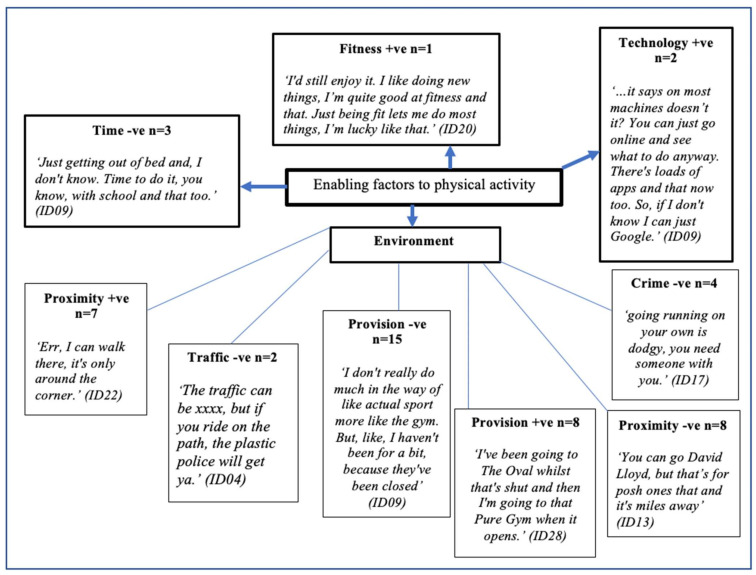
Enabling factors to physical activity. +ve = positive; −ve = negative.

**Figure 5 ijerph-18-04923-f005:**
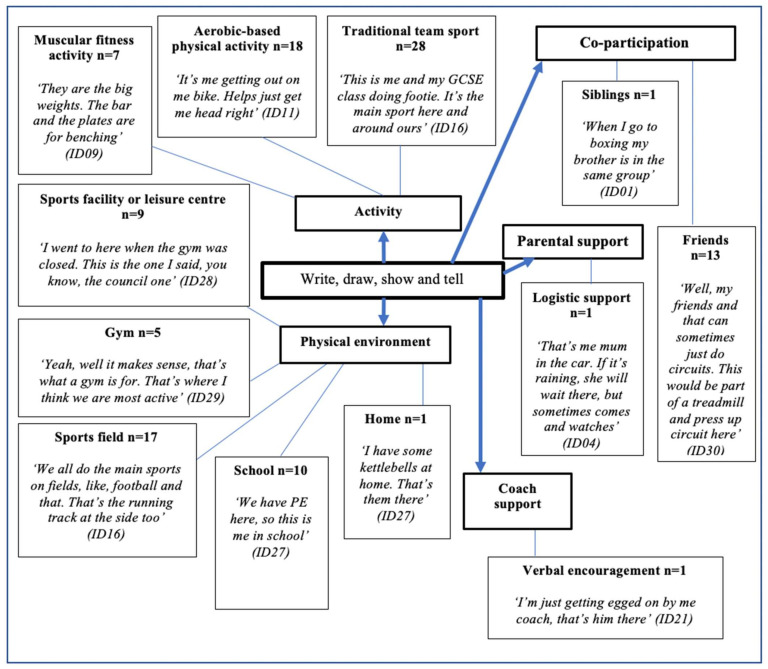
Write, draw, show, and tell. +ve = positive; −ve = negative.

**Figure 6 ijerph-18-04923-f006:**
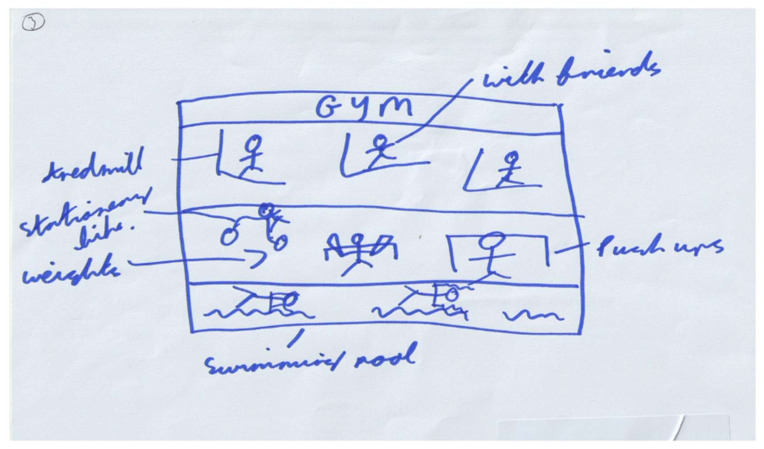
Drawing illustrating co-participation in physical activity with friends. (ID30). *“You can like, use the gym here (school), but you probably couldn’t do it with your mates and that. They (teachers) don’t trust us I don’t think. In a gym is sound, no teachers so you can just train properly.”*.

**Figure 7 ijerph-18-04923-f007:**
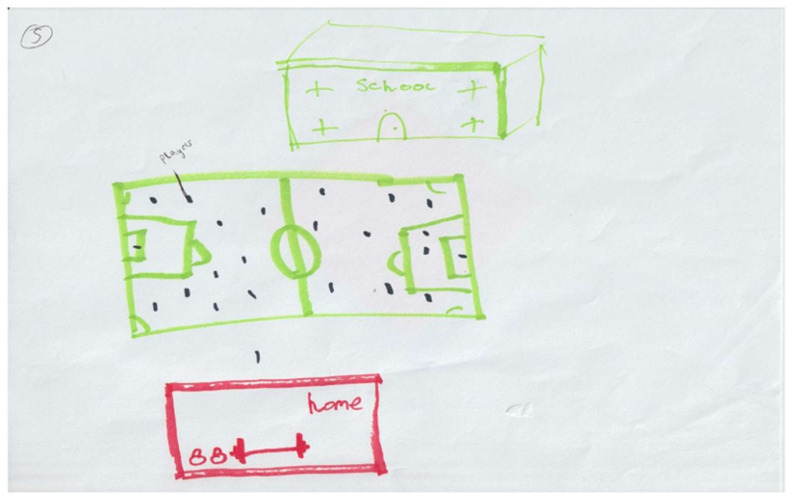
Drawing illustrating the perceived difference in activities conducted at school and out of school. (ID27). *“School is mostly football and that. When I go home I can do weights, it’s just not something we do here [school]”.*

**Table 1 ijerph-18-04923-t001:** Example write, draw, show, and tell questions.

Topic	Question
Physical Activity Knowledge	How much physical activity should we do each week? What is the difference between aerobic and resistance activity?
Predisposing	What physical activities do you take part in at school and out of school? Why do you do more of this activity than others?
Enabling	What opportunities do you have to participate in muscular fitness activity?
Reinforcing	When you are physically active, who do you do these activities with? What do parents, friends, coaches, or teachers think of you doing muscular fitness activity?

**Table 2 ijerph-18-04923-t002:** Top 10 responses for the icebreaker activity.

Word	Count
fun	18
enjoyable	11
healthy	9
active	7
exercise	4
movement	4
physical	4
running	4
energetic	3
exciting	3
fitness	3
hard	3
interesting	3
sport	3
alive	2

## Data Availability

Participants in this study were under 18 years of age and supporting data for this study is not openly available as participants did not provide informed consent for data sharing.

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
