# Peer review of "“It’s Just Not Something We Do at School”. Adolescent Boys’ Understanding, Perceptions, and Experiences of Muscular Fitness Activity"

_ijerph, 2021, doi:10.3390/ijerph18094923_

Round 1

Reviewer 1 Report

The authors present an interesting article, compliant with the journal's topic.
The study has a great strength in that it is contextualized in the adolescents' environment. The approach, through a qualitative methodology, uses different data collection techniques (focus interviews, sticky notes, drawings,...) that allow us to collect the perceptions, feelings, and emotions of young people about muscular fitness activity.
Below I make some suggestions to the authors to improve the understanding of the procedures, and to know about practical applications of the study:
Method: Please report on:
how many researchers were involved in the coding?
How was the reliability of the agreements on the choice of codes assessed?
Has any training and testing for intra-observer and inter-observer reliability been conducted?
The discussion of the results is thorough and adequate. I congratulate the authors.
Practical applications:
Adolescents attribute the school as a facilitator of physical activity knowledge. The discussion values the school context for facilitating knowledge about muscular physical activity. It is important to provide some practical guidance:
Could you make any suggestions for practical applications to the development of knowledge surrounding muscular fitness and for the development of muscular fitness in the school setting?
Could you make some suggestions on how to extend the social network both outside and inside the school to support the development of muscular fitness?

Limitations: In the experience of the researchers, was the length of the interview sufficient, or was it a limitation (20 minutes)?

Author Response

Reviewer 1 responses

The authors present an interesting article, compliant with the journal's topic.
The study has a great strength in that it is contextualized in the adolescents' environment. The approach, through a qualitative methodology, uses different data collection techniques (focus interviews, sticky notes, drawings,...) that allow us to collect the perceptions, feelings, and emotions of young people about muscular fitness activity.

The authors would like to thank the reviewers for their comments. We have taken time to consider and address all comments indicated in the review reports, and as a result believe that the revised version is much improved in line with the journal publication requirements. All revisions to the manuscript are highlighted yellow.

Below I make some suggestions to the authors to improve the understanding of the procedures, and to know about practical applications of the study:

Method: Please report on:

how many researchers were involved in the coding?

Two authors (1&3) agreed upon the coding protocol and strategy and ensured that the coding terminology was in line with the available literature investigating physical activity. Further review took the form of a presentation of the verbatim quotations and drawings to the third author as a critical friend who had previously independently reviewed the data sources and cross-examined the data sources against the themes in reverse to offer alternative perspectives. This process was repeated until a minimum 90% agreement level had been reached by the two authors. Methodological rigour, credibility and transferability were achieved via verbatim transcription of data and triangular consensus procedures, and comparison of pen profiles with verbatim and illustration data accentuated dependability. This detail has now been added to lines 180-188 in addition to the existing contribution statement at the end of the manuscript.

How was the reliability of the agreements on the choice of codes assessed?

Authors 1&3 conducted multiple meetings to ensure agreement in the coding. Where there was disagreement, the authors discussed and amended the coding process after agreeing the choice of code. We have amended the manuscript to demonstrate the agreement between authors and to provide insight into how many researchers were involved in the coding process as commented on above. These amendments can be found at lines 180-188. The authors thank reviewer 1 for highlighting this area for amendment and believe the addition brings further clarity to the already thorough methodology used in this body of work.

Has any training and testing for intra-observer and inter-observer reliability been conducted?

The authors have published previous qualitative research using the same methodology and data analyses procedures employed in the present study (MacKintosh, Knowles, Ridgers, & Fairclough, 2011; Noonan, Boddy, Fairclough, & Knowles, 2016; Taylor, Noonan, Knowles, Owen, & Fairclough, 2018). Data management and coding was individually conducted and the process was then repeated until a minimum 90% agreement level had been reached by the two authors and in agreement with inter-rater reliability suggestions (McAlister et al., 2017; O’Connor & Joffe, 2020; Roberts, Dowell, & Nie, 2019). These details have been included in lines 185-188.

The discussion of the results is thorough and adequate. I congratulate the authors.

The authors thank the reviewer for the positive feedback and appreciate the acknowledgement of the work conducted by the researchers involved.

Practical applications:
Adolescents attribute the school as a facilitator of physical activity knowledge. The discussion values the school context for facilitating knowledge about muscular physical activity. It is important to provide some practical guidance:
Could you make any suggestions for practical applications to the development of knowledge surrounding muscular fitness and for the development of muscular fitness in the school setting? Could you make some suggestions on how to extend the social network both outside and inside the school to support the development of muscular fitness?

The authors thank the reviewer for raising this. Although recommendations are touched upon within the discussion, we have now explicitly stated some areas for future research at lines 497-510. We believe that this additional information provides clarity on the direction of future research.

Limitations: In the experience of the researchers, was the length of the interview sufficient, or was it a limitation (20 minutes)?

The rationale surrounding the 20 minute length was based on data saturation, limitations set by the school and taking into account previous research. During the 20 minute focus group interviews there were emerging themes being repeated and justifying the saturation, focus group interviews conducted any longer may not have elicited any further themes(Krueger & Casey, 2015; Parker et al., 2012; Wong, 2008). Furthermore, the use of concurrent activity through the use of write, draw and show activities will have artificially extended the focus group interview length.

Due to the time constraints placed upon the research team by the school, the first two focus group interviews were 20 minutes in length. In the interest of consistency, we mirrored this time across further focus group interviews to ensure standardisation. Finally, the 20 minute focus group/group interview length has been used in previous research involving young people( Davies & Robinson, 2010) and has been highlighted as an appropriate time length in a recent integrative literature review (Adler, Salanterä, & Zumstein-Shaha, 2019). However, given that this has been highlighted by both reviewers and that it might be perceived as being shorter than in some other studies, we have highlighted this aspect of the study as a potential limitation in the latter section of the manuscript on lines 480-496.

Adler, K., Salanterä, S., & Zumstein-Shaha, M. (2019). Focus Group Interviews in Child, Youth, and Parent Research: An Integrative Literature Review. International Journal of Qualitative Methods, 18. https://doi.org/10.1177/1609406919887274

Davies, C., & Robinson, K. (2010). Hatching babies and stork deliveries: Risk and regulation in the construction of children’s sexual knowledge. Contemporary Issues in Early Childhood. https://doi.org/10.2304/ciec.2010.11.3.249

Krueger, R. A., & Casey, M. A. (2015). Focus groups: A practical guide for applied research 5th Edition. Focus Groups: A Practical Guide for Applied Research.

McAlister, A. M., Lee, D. M., Ehlert, K. M., Kajfez, R. L., Faber, C. J., & Kennedy, M. S. (2017). Qualitative coding: An approach to assess inter-rater reliability. In ASEE Annual Conference and Exposition, Conference Proceedings. https://doi.org/10.18260/1-2--28777

O’Connor, C., & Joffe, H. (2020). Intercoder Reliability in Qualitative Research: Debates and Practical Guidelines. International Journal of Qualitative Methods, 19, 160940691989922. https://doi.org/10.1177/1609406919899220

Parker, A. E., Halberstadt, A. G., Dunsmore, J. C., Townley, G., Bryant, A., Thompson, J. A., & Beale, K. S. (2012). Emotions are a window into one’s heart”: a qualitative analysis of parental beliefs about children’s emotions across three ethnic groups. Monographs of the Society for Research in Child Development. https://doi.org/10.1111/j.1540-5834.2012.00676.x

Roberts, K., Dowell, A., & Nie, J. B. (2019). Attempting rigour and replicability in thematic analysis of qualitative research data; A case study of codebook development. BMC Medical Research Methodology, 19(1). https://doi.org/10.1186/s12874-019-0707-y

Wong, L. P. (2008). Focus group discussion: A tool for health and medical research. Singapore Medical Journal.

Reviewer 2 Report

Dear authors.

The manuscript is of great scientific quality and has been made following the usual guidelines of qualitative research. I give you my comments in case they could be used to improve the paper.

I like the title you have chosen.

Introduction:

I wonder if girls are not interested in muscle exercise in the UK. You should better justify the selection of the target population based solely on adolescent boys.

I find it interesting that you justify the research method used. It is remarkable that you did a triangulation using drawing techniques (although I think these are more suitable in younger children, where oral or written expression is not as well developed). However (the subject of study is not very thorny to justify using techniques beyond the interview or the focus group) it is adequate and gives greater validity to the results.

Methods.

You must give details of the type of research design used. I mean, it is not enough to say that it was a qualitative study based on focus groups. It is necessary to indicate the theoretical-methodological orientation with which the study is framed (was it a phenomenological, ethnographic study, or perhaps based on grounded theory?).

In the participants section, you must expand the information on the sample (mean age, for example) and detail the type of sampling used, as well as justify the inclusion / exclusion criteria (previous physical condition, obesity, etc.) that could influence student opinions.

In the explanation of the procedure, you must indicate the number of participants in each focus group and the total number of groups carried out. To report the results and write the article I recommend that you use and cite the COREQ checklist (COnsolidated criteria for REporting Qualitative research) Tong A, Sainsbury P, Craig J. Consolidated criteria for reporting qualitative research (COREQ): a 32-item check list for interviews and focus groups. International Journal for Quality in Health Care. 2007. Volume 19, Number 6: pp. 349 – 357

I wonder if 20 minutes was enough time to carry out the icebreaker dynamic, the presentation of the questions and for all the participants to write, draw and speak ... In qualitative research, focus groups usually take longer to develop.

Results.

I realize that the first result does not serve to answer the objectives of the study, but rather measures the children's impression of the initial icebreaker dynamics.

The figures represent concept maps that help a lot to understand the results.

Lines 196-197. You have cited in APA style. Citation required in Vancouver style.

Discussion.

The first paragraph once again expresses the objectives of the study that were already described at the end of the introduction. This can be repetitive.

The discussion is deep and rich; however, it gives me the feeling that aspects that should be included in the results have been included in the discussion section. For example, when presenting the children's drawings or when you present results that have not been previously reported in the results section. I think the discussion section could be better understood if you clean up the results and simply explain and interpret the main findings of the study.

A section of limitations should be included at the end of the discussion. It is important to state the main limitations of the study. It may help you to keep in mind the limitations of your study noted by the reviewers.

Author Response

Reviewer 2 responses

The manuscript is of great scientific quality and has been made following the usual guidelines of qualitative research. I give you my comments in case they could be used to improve the paper.

I like the title you have chosen.

The authors would like to thank the reviewers for their comments. We have taken time to consider and address all comments indicated in the review reports, and as a result believe that the revised version is much improved in line with the journal publication requirements. All revisions to the manuscript are highlighted yellow.

Introduction:

I wonder if girls are not interested in muscle exercise in the UK. You should better justify the selection of the target population based solely on adolescent boys.

The reviewer raises a good point, and this is certainly an area for future research. However, in this instance we selected the target population based on the recognised desire of boys to participate in, and proven efficacy of, muscular fitness activity in adolescent boys. However, much of the previous literature that is used to draw these conclusions is quantitative in nature and we were keen to provide a qualitative insight into adolescent boys’ understanding, perceptions and experiences of physical activity and the role muscular fitness has within boys’ physically active lifestyles to advance the current evidence base.

I find it interesting that you justify the research method used. It is remarkable that you did a triangulation using drawing techniques (although I think these are more suitable in younger children, where oral or written expression is not as well developed). However (the subject of study is not very thorny to justify using techniques beyond the interview or the focus group) it is adequate and gives greater validity to the results.

We wanted to provide a mixture of communication methods to ensure all adolescents could participate, and the chosen methods were suitable for this purpose. The combination of methods provided the participants alternative ways of expression and enabled a deeper exploration of their understanding, perceptions and experiences by not limiting them to verbal communication. It was envisioned that the interactive and dual methods-based approach would foster greater inclusivity and would elicit more representative and detailed perceptions on physical activity and muscular fitness that perhaps would remain uncovered when using traditional singular methods-based approaches.

Methods. 

You must give details of the type of research design used. I mean, it is not enough to say that it was a qualitative study based on focus groups. It is necessary to indicate the theoretical-methodological orientation with which the study is framed (was it a phenomenological, ethnographic study, or perhaps based on grounded theory?).

The authors thank the reviewer for highlighting this. We have now added in the research design used and included details within the manuscript at line 99.

In the participants section, you must expand the information on the sample (mean age, for example) and detail the type of sampling used, as well as justify the inclusion / exclusion criteria (previous physical condition, obesity, etc.) that could influence student opinions.

This information has now been added at lines 92-94, 103-104.

In the explanation of the procedure, you must indicate the number of participants in each focus group and the total number of groups carried out. To report the results and write the article I recommend that you use and cite the COREQ checklist (COnsolidated criteria for REporting Qualitative research) Tong A, Sainsbury P, Craig J. Consolidated criteria for reporting qualitative research (COREQ): a 32-item check list for interviews and focus groups. International Journal for Quality in Health Care. 2007. Volume 19, Number 6: pp. 349 – 357

Thank you for raising this point. After reading the paper we have made some amendments in our reporting, many of which you have highlighted in the review process such as design and participant characteristics. We have now explicitly stated we used the checklist to ensure our reporting is thorough, lines 85-86. Thank you again for raising this, I am sure this will also come in useful in future qualitative studies conducted by the authors.

I wonder if 20 minutes was enough time to carry out the icebreaker dynamic, the presentation of the questions and for all the participants to write, draw and speak ... In qualitative research, focus groups usually take longer to develop.

The rationale surrounding the 20 minute length was based on data saturation, limitations set by the school and taking into account previous research. During the 20 minute focus group interviews there were emerging themes being repeated and justifying the saturation, focus group interviews conducted any longer may not have elicited any further themes (Krueger & Casey, 2015; Parker et al., 2012; Wong, 2008). Furthermore, the use of concurrent activity through the use of write, draw and show activities will have artificially extended the focus group interview length.

Due to the time constraints placed upon the research team by the school, the first two focus group interviews were 20 minutes in length. In the interest of consistency, we mirrored this time across further focus group interviews to ensure standardisation. Finally, the 20 minute focus group/group interview length has been used in previous research involving young people (Davies & Robinson, 2010) and has been highlighted as an appropriate time length in a recent integrative literature review study (Adler, Salanterä, & Zumstein-Shaha, 2019). However, given that this has been highlighted by both reviewers, we have decided to highlight this aspect of the study as a potential limitation in the latter section of the manuscript on lines 480-496.

Results.

I realize that the first result does not serve to answer the objectives of the study, but rather measures the children's impression of the initial icebreaker dynamics.

This task was used as a familiarisation to build rapport and understanding of physical activity.

The figures represent concept maps that help a lot to understand the results.

Our decision to use the term pen profiles was informed by previous qualitative research involving children and young people (Knowles, Parnell, Stratton, & Ridgers, 2013; MacKintosh, Knowles, Ridgers, & Fairclough, 2011; Noonan, Boddy, Fairclough, & Knowles, 2016; Taylor, Noonan, Knowles, Owen, & Fairclough, 2018).

Lines 196-197. You have cited in APA style. Citation required in Vancouver style.

This has now been rectified using our reference manager and should read in the correct format whilst providing the reference for Welk (1999). Thank you.

Discussion.

The first paragraph once again expresses the objectives of the study that were already described at the end of the introduction. This can be repetitive.

Given the wealth of data presented in the results section, we feel that it is important to inform the reader of the study’s objectives and explicitly state how these objectives were met. We feel that this information is of benefit to the reader as it sets the scene for the critical evaluation that follows.

The discussion is deep and rich; however, it gives me the feeling that aspects that should be included in the results have been included in the discussion section. For example, when presenting the children's drawings or when you present results that have not been previously reported in the results section. I think the discussion section could be better understood if you clean up the results and simply explain and interpret the main findings of the study.

The authors believe that the data is presented in a manner that is not overwhelming and figure 5 covers the visual data captured by the write draw show and tell activity. Coding the visual data and placing into pen-profiles allowed the authors to align the findings with recognised themes and build upon the expansive physical activity research currently available. The visual representations provided in the discussion support the pen-profiles (Figure 5) and provide the reader with insight into the visual data provided, further supporting the discussion points presented. Finally, we would like to keep the example visual data in the discussion as we believe the presentation of visual data supports what the authors have concluded and further enhances the credibility and dependability of our findings whilst supporting our discussion points. Consistent with previous studies adopting the pen-profile approach (Knowles et al., 2013; MacKintosh et al., 2011; Noonan et al., 2016; Taylor et al., 2018), the present study used the results section to explicitly make known to the reader the emerging higher order themes. The discussion section was then used to add context to the individual pen-profiles. This approach minimises replication of information. 

A section of limitations should be included at the end of the discussion. It is important to state the main limitations of the study. It may help you to keep in mind the limitations of your study noted by the reviewers.

Supported by the limitations noted by the reviewers, we have now acknowledged them within the manuscript and report as such at the end of the discussion. The limitations can be found at lines 480-496.

Adler, K., Salanterä, S., & Zumstein-Shaha, M. (2019). Focus Group Interviews in Child, Youth, and Parent Research: An Integrative Literature Review. International Journal of Qualitative Methods, 18. https://doi.org/10.1177/1609406919887274

Davies, C., & Robinson, K. (2010). Hatching babies and stork deliveries: Risk and regulation in the construction of children’s sexual knowledge. Contemporary Issues in Early Childhood. https://doi.org/10.2304/ciec.2010.11.3.249

Knowles, Z. R. ebecc., Parnell, D., Stratton, G., & Ridgers, N. D. ian. (2013). Learning from the experts: exploring playground experience and activities using a write and draw technique. Journal of Physical Activity & Health. https://doi.org/10.1123/jpah.10.3.406

Krueger, R. A., & Casey, M. A. (2015). Focus groups: A practical guide for applied research 5th Edition. Focus Groups: A Practical Guide for Applied Research.

MacKintosh, K. A., Knowles, Z. R., Ridgers, N. D., & Fairclough, S. J. (2011). Using formative research to develop CHANGE!: A curriculum-based physical activity promoting intervention. BMC Public Health. https://doi.org/10.1186/1471-2458-11-831

Noonan, R. J., Boddy, L. M., Fairclough, S. J., & Knowles, Z. R. (2016). Write, draw, show, and tell: A child-centred dual methodology to explore perceptions of out-of-school physical activity. BMC Public Health.

Parker, A. E., Halberstadt, A. G., Dunsmore, J. C., Townley, G., Bryant, A., Thompson, J. A., & Beale, K. S. (2012). Emotions are a window into one’s heart”: a qualitative analysis of parental beliefs about children’s emotions across three ethnic groups. Monographs of the Society for Research in Child Development. https://doi.org/10.1111/j.1540-5834.2012.00676.x

Taylor, S. L., Noonan, R. J., Knowles, Z. R., Owen, M. B., & Fairclough, S. J. (2018). Process evaluation of a pilot multi-component physical activity intervention - Active schools: Skelmersdale. BMC Public Health, 18(1), 1383. https://doi.org/10.1186/s12889-018-6272-1

Wong, L. P. (2008). Focus group discussion: A tool for health and medical research. Singapore Medical Journal.
